# Large linear non-saturating magnetoresistance and high mobility in ferromagnetic MnBi

Yangkun He [1✉], Jacob Gayles [1,2], Mengyu Yao [1], Toni Helm [1,3], Tommy Reimann [3], Vladimir N. Strocov [4], Walter Schnelle[1], Michael Nicklas [1], Yan Sun [1], Gerhard H. Fecher[1] & Claudia Felser[1]

A large non-saturating magnetoresistance has been observed in several nonmagnetic topological Weyl semi-metals with high mobility of charge carriers at the Fermi energy. However, ferromagnetic systems rarely display a large magnetoresistance because of localized electrons in heavy $d$ bands with a low Fermi velocity. Here, we report a large linear non-saturating magnetoresistance and high mobility in ferromagnetic MnBi. MnBi, unlike conventional ferromagnets, exhibits a large linear non-saturating magnetoresistance of 5000% under a pulsed field of 70 T. The electrons and holes' mobilities are both 5000 cm$^2$V$^{-1}$s$^{-1}$ at 2 K, which are one of the highest for ferromagnetic materials. These phenomena are due to the spin-polarised Bi 6$p$ band's sharp dispersion with a small effective mass. Our study provides an approach to achieve high mobility in ferromagnetic systems with a high Curie temperature, which is advantageous for topological spintronics.

[1] Max-Planck-Institute for Chemical Physics of Solids, Dresden, Germany. [2] Department of Physics, University of South Florida, Tampa, FL, USA. [3] Dresden High Magnetic Field Laboratory (HLD-EMFL), Helmholtz-Zentrum Dresden–Rossendorf, Dresden, Germany. [4] Swiss Light Source, Paul Scherrer Institut, Villigen, Switzerland. ✉email: yangkun.he@cpfs.mpg.de

Weyl fermions, massless chiral quasiparticles in the momentum space, host some of the most intriguing transport phenomena, such as the anomalous Hall effect (AHE)[1], anomalous Nernst effect[2], (quantum) oscillations[3,4], or a large positive transverse magnetoresistance (MR)[5,6]. Nonmagnetic examples of topological Weyl semi-metals, such as TaAs[7], NbP[6], and WTe$_2$[5], have been reported with a giant positive MR of more than $10^5$% at low temperatures. Its origin lies in the massless Weyl states associated to the linear band crossings at the Fermi energy and a high Fermi velocity. Magnetic topological semimetals allow for the manipulation of the Fermi surface topology by external electromagnetic fields and are therefore fundamental in topological spintronics research.

Only a few examples of ferromagnetic topological materials, such as Fe$_3$Sn$_2$[8], Co$_3$Sn$_2$S$_2$[1], and Co$_2$MnGa[2,9], have been reported to exhibit a weakly positive MR of 20–30%. Although these values are much larger than those of their non-topological ferromagnetic relatives, they are significantly lower than those of the leading nonmagnetic examples (see Fig. 1e). This is because the mobility $\mu$ of ferromagnetic Weyl semi-metals is still considerably less than $10^3\,\mathrm{cm^2\,V^{-1}\,s^{-1}}$, due to both electron and spin scattering. Only in the case of higher mobilities is the criterion $\mu B > 1$ fulfilled ($B$ is the applied magnetic field), enabling the observation of quantum effects and thus a large linear non-saturating MR[10].

Giant MR has attracted a broad interest because of its theoretical aspects and its commercial applications, such as spin-valve sensors[11] and hard disc read/write heads[12]. In ordinary nonmagnetic metals, the MR is usually a small value of only a few percent. In ferromagnetic materials, which are metals in most cases, the MR is generally small and negative (inset of Fig. 1e) owing to spin disorder scattering. Considerable efforts have been made to find new FM materials with a large MR. A giant MR of 150% was observed in Fe–Cr thin-film heterostructures at 4.2 K, associated with spin-flip scattering[13]. In some ferromagnetic semiconductors, such as HgCr$_2$Se$_4$[14], LaMnO$_3$[15], and EuO[16], a colossal MR was reported near the Curie temperature ($T_c$), which is due to a magnetic-field-induced semiconductor-to-metal phase transition. However, this phase transition requires a large magnetic field and is usually accompanied by a large hysteresis[17]. Yet,

Weyl semimetals have more simplistic and predictable electronic structures. Due to the topology—the electronic structure suppresses backscattering and spin scattering which leads to some of the largest MR values found.

Herein, we report the transport properties of single-crystal ferromagnetic MnBi. A large linear non-saturating MR, defined as the ratio of the change in resistance $\rho(B)$ in response to an applied magnetic field $[\rho(B) - \rho(0)]/\rho(0)$. Furthermore, we find one of the largest mobilities in ferromagnetic materials. These phenomena are due to the linear dispersion of the $6p$ band at the Fermi energy with a small effective mass as confirmed from state-of-the-art first-principle calculations, as well as angle-resolved photoemission spectroscopy (ARPES) and Shubnikov–de Haas (SdH) oscillations experiments on single crystals. In this sense, MnBi with an MR comparable to nonmagnetic semimetals is unique in comparison to the topological ferromagnetic counterparts. And a giant MR single crystal device that is tunable purely by the magnetic moment can have far-reaching implications for spintronic devices.

## Results

**Crystal and magnetic structure.** MnBi crystallizes in a NiAs-type crystal structure with the space group $P6_3/mmc$ (No. 194). It consists of alternating Mn and Bi layers, as shown in Fig. 1a. The lattice constants are obtained as $a = 4.2876(5)$ Å and $c = 6.1154(5)$ Å, based on X-ray diffraction results. Known as a room-temperature hard magnet with a large maximum energy product[18], MnBi exhibits a large moment of 3.94 $\mu_B$ per Mn atom and a high Curie temperature ($T_c > 630$ K)[19]. Below the spin-reorientation transition temperature $T_{SR1} = 137$ K, the moment gradually rotates away from the $c$-axis upon cooling[20] and abruptly flips close to the $ab$-plane[19] at $T_{SR2} = 87$ K.

The temperature-dependent longitudinal resistivity of MnBi is shown in Fig. 1b. The high-quality single crystal grown via the flux method shows a metallic behavior with a sizeable residual-resistivity ratio (RRR) of 100 (see also Supplementary Fig. 1 in the supplemental material). This indicates the high quality of the single crystal and very high mobility, suppressed by phonon

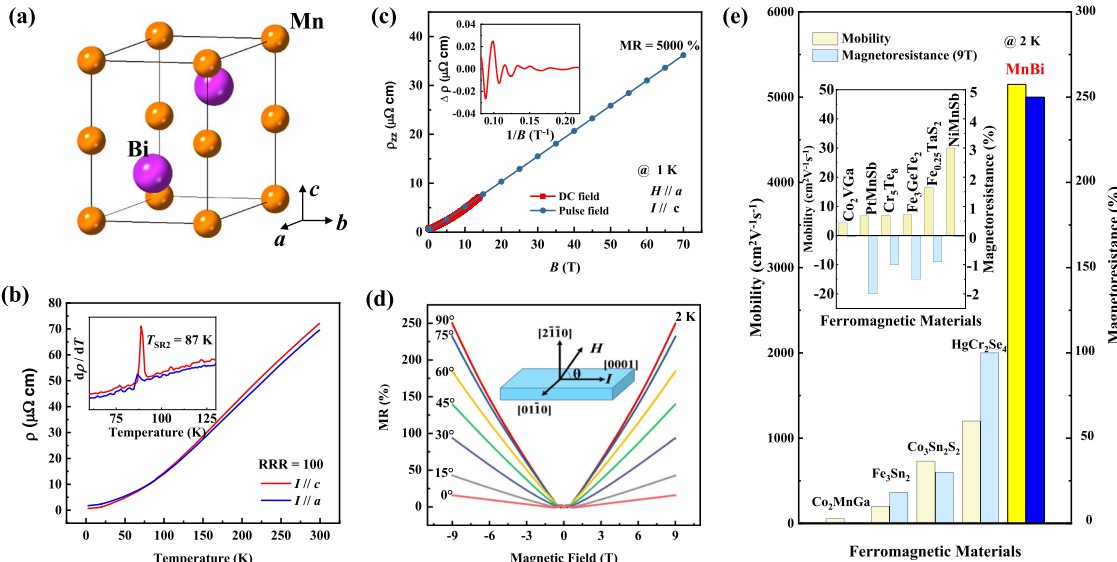

**Fig. 1 Crystal structure and measured resistivity of MnBi. a** The crystal structure of MnBi in space group $P6_3/mmc$. **b** Longitudinal resistivity along with the $c$- and $a$-axes. The inset shows a kink for d$\rho$/d$T$ at $T_{SR2}$. **c** Non-saturating linear MR under a pulse-field of 70 T at 1 K. The inset shows the SdH oscillations obtained from the resistivity data below 14 T under a DC magnetic field. **d** Angular-dependent MR at 2 K. **e** Comparison of its MR and high mobility with those of other ferromagnetic materials. Data for other ferromagnetic materials come from refs. [1,8,9,41–45]. The inset shows the low mobility and negative MR for ordinary ferromagnetic materials.

scattering during heating[21]. The longitudinal resistivity $\rho$ decreases to 0.69 μΩ cm along the $c$-axis and 1.7 μΩ cm at 2 K along the $a$-axis. An additional kink in the differential resistivity at a $T_{SR2}$ of 87 K is shown in the inset. Strictly speaking, the crystal structure is orthorhombic below $T_{SR2}$ due to the magnetostriction[19], but this small distortion is ignored in the analysis.

**Magnetoresistance.** A non-saturating linear MR of 5000% under a pulsed-field of 70 T at 1 K is shown in Fig. 1c. The data measured under a direct current (DC) magnetic field of 14 T are also shown for comparison. The Shubnikov–de Haas oscillations were observed with a single frequency at 23 T by subtracting a cubic polynomial from the resistivity data below 14 T, as shown in the inset. The detailed data are shown in Supplementary Fig. 2 of the supplement. The MR is strongly angular dependent, as shown in Fig. 1d. The MR decreases from 250 to 15% at 9 T when the field changes from perpendicular to parallel to the current.

With increasing temperature, the MR becomes less evident in Fig. 2a, and it has a small negative value above 90 K, just as a normal ferromagnet (see Fig. 2b). The non-saturating MR reveals a two-charge carrier behavior at low temperatures, where holes and electrons have almost the same density[5]. The individual quantum levels associated with the electron orbits should be distinct to realize this phenomenon[10]. In other words, our results indicate that there exists an approximately linear energy spectrum in the band structure and carriers of a very low effective mass.

The MR was also measured under different fields and current directions. It is positive at low temperatures, independent of the field and current directions. The MR is the largest when $I//c$, and it is lower when the current is in the $ab$ plane. The larger and non-saturated MR when $I//c$ may result from the more effective compensation of electrons and holes. Above 90 K, the MR is negative during magnetization when $H//a$ and $I//c$ or when $H//c$ and $I//a$ (Fig. 2b, f), whereas it is positive when both the current and field are in-plane (Fig. 2d). Notably, a butterfly shape is observed in Fig. 2f, owing to the hysteresis at high temperatures when MnBi becomes a hard magnet in a micron-sized single crystal.

**Hall resistivity and two-charge-carrier behavior.** The Hall resistivity $\rho_H$ at different temperatures is shown in Fig. 3a when the field is parallel to the $a$-axis, and the current is parallel to the $c$-axis. At 2 K, the curve further confirms the two-charge-carrier behavior at low temperatures. Note that the curves correspond to the ordinary Hall effect rather than the AHE. The reason for rejecting an anomalous contribution is that the magnetization, which determines the AHE field dependence, is already saturated at 0.5 T. In contrast, the observed Hall signal does not become saturated, as shown in the inset.

There is no suitable two-charge-carrier model to fit the MR of a ferromagnetic material because the magnetization process removes the domain walls and decreases the domain wall scattering. This leads to an additional negative MR, as shown in Fig. 2a when the field is less than 0.5 T. However, for the Hall signal, the anomalous Hall resistivity is zero without a magnetic field owing to:

1. the negligible hysteresis due to the easy plane magnetic structure at a low temperature and
2. the intrinsic anomalous Hall resistivity that is two orders lower than the normal Hall resistivity at saturation because of the extremely small longitudinal $\rho$ ($\sigma_H = -\rho_H/\rho^2$, where $\sigma_H$ is the intrinsic Hall conductivity deduced from the slope in Fig. 3d), which is similar to that of pure Fe[22] and Gd[23].

Therefore, the AHE is negligible compared with the ordinary Hall effect at low temperatures. We fit the two-charge-carrier model using Hall resistivity as follows:

$$\rho_H = -\frac{1}{e}\frac{(n\mu^2 - p\mu'^2) - (p-n)\mu^2\mu'^2 B^2}{(n\mu + p\mu')^2 + [(p-n)\mu\mu' B]^2}B \quad (1)$$

where $p$ and $n$ are the charge carrier densities for holes and electrons, respectively, and $\mu'$ and $\mu$ are the mobilities of holes and electrons. The fitted curve at 2 K is shown in the inset of Fig. 3a. The densities of both holes and electrons are almost the same at approximately $9 \times 10^{20}$ cm$^{-3}$ (0.02 hole/electron per formula), and they are temperature independent, consistent with previously reported values[24]. The mobilities of both the charge carriers are high, i.e., approximately 5000 cm$^2$ V$^{-1}$ s$^{-1}$ at 2 K. This value

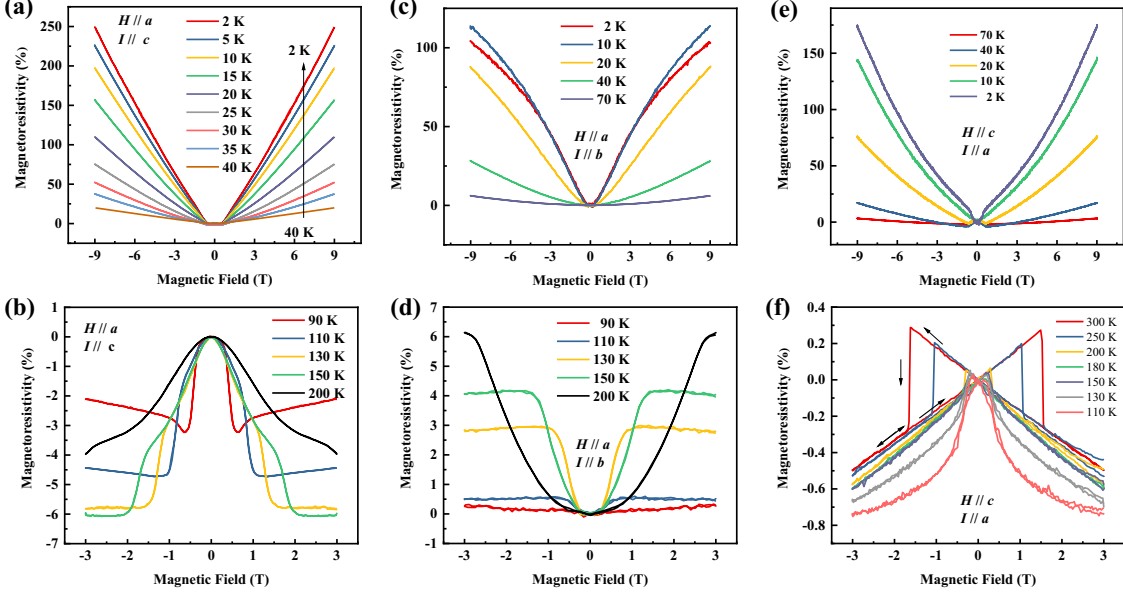

**Fig. 2 MR under different fields and current directions. a**, **b** $H//a$ and $I//c$. **c**, **d** $H//a$ and $I//b$. **e**, **f** $H//c$ and $I//a$. The data in (**e**) and (**f**) are obtained using a micron-sized single crystal cut using the focused ion beam technique because of the bar shape of the crystal, whereas the data in a-d are obtained by using bulk single crystals.

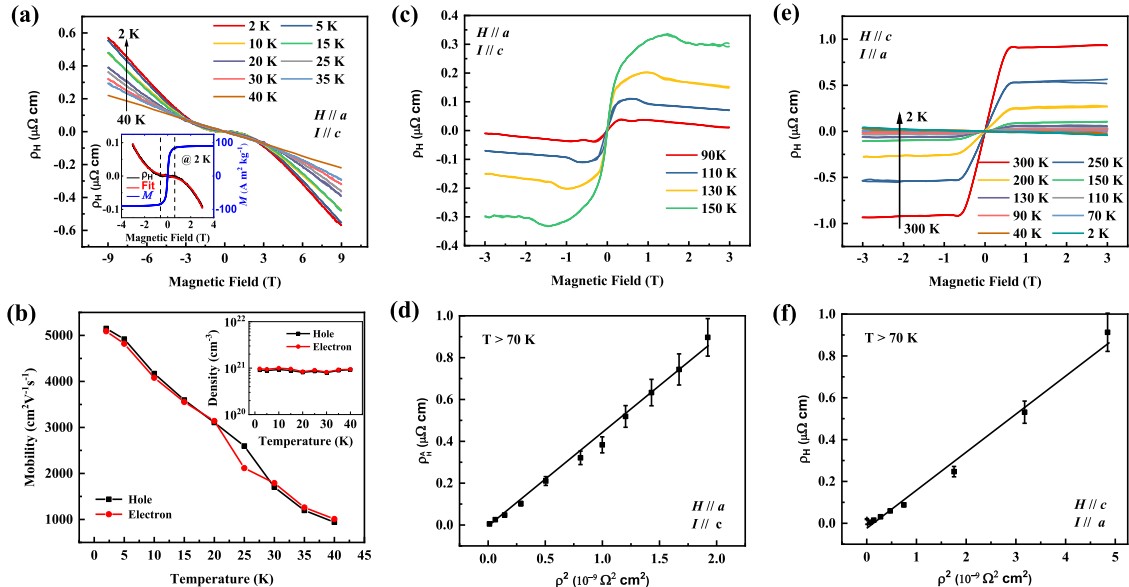

**Fig. 3 Hall effect of MnBi. a–d** for $H//a$ and $I//c$, and **e–f** for $H//c$ and $I//a$. **a** Hall effect between 2 and 40 K with typical two-charge-carrier features. The inset shows the magnetization and fits using a two-charge-carrier model. The field-dependent AHE, proportional to the magnetization, is saturated at approximately 0.5 T shown by dashed lines. In contrast, the Hall resistivity slope still changes under a magnetic field greater than 4 T due to the different charge carriers. **b**, Mobility and charge carrier density (inset) at different temperatures. **c, e** Hall resistivity at different temperatures. **d, f** Anomalous Hall resistivity versus the longitudinal resistivity square. The slopes of these curves are the intrinsic anomalous Hall conductivity. The error bar due to the uncertainty of the geometry of the sample is estimated as 10%.

decreases with increasing temperature to 900 cm$^2$ V$^{-1}$ s$^{-1}$ at 40 K. Fitting with three or more charge carrier models would be more accurate because of the complicated band structure at the Fermi energy, but the conclusion of the high mobility does not change.

We compared the MR and mobility of MnBi with those of other ferromagnetic materials at 2 K, as shown in Fig. 1e. Ordinary ferromagnets usually have a negative MR of a few percent and rather low mobility ($< 50$ cm$^2$ V$^{-1}$ s$^{-1}$) due to the large effective mass of $d$ electrons and spin scattering. Topological materials show a large positive MR with high mobilities due to the topological bands and small effective masses. MnBi has the highest mobility among ferromagnetic materials.

At temperatures above 90 K, the AHE becomes noticeable, as shown in Fig. 3c, because the longitudinal resistivity is much larger. At high temperatures, the conductivity is in the range of $10^4$–$10^6$ $\Omega^{-1}$ cm$^{-1}$, where the intrinsic Berry phase dominates the Hall effect[25]. The anomalous Hall resistivity versus the longitudinal resistivity square is shown in Fig. 3d. A linear fit provides an intrinsic $\sigma_H$ of $-450$ $\Omega^{-1}$ cm$^{-1}$, comparable to $-640$ $\Omega^{-1}$ cm$^{-1}$ from the Berry curvature calculation (see Fig. 4b). More information about the Hall can be found in Supplementary Fig. 3. The anomalous Hall resistivity when the field is along the $c$-axis and current long the $a$-axis is shown in Fig. 3e. The intrinsic $\sigma_H$ is smaller as $-187$ $\Omega^{-1}$ cm$^{-1}$, as shown in Fig. 3f.

**Band structure**. The calculated band structure with a moment parallel to the $a$-axis is shown in Fig. 4a. It presents a typical semi-metallic band structure with both electron and hole pockets. The band structure was confirmed using ARPES. Based on the photon energy ($h$) dependent measurement, $k_z = 0$ plane can be reached by using photon energy $h\nu = 400$ eV. The Fermi surface in the plane perpendicular to the $z$-direction when $k_z = 0$ exhibits a flower-like texture, as shown in Fig. 4e. The calculated Fermi surface is shown in Fig. 4f and Supplementary Fig. 4. A series of hole pockets located at the $\Gamma$ point is surrounded by six electron

pockets located at the M points, confirming the two-charge-carrier behavior in the transport measurements. Note that the X-ray beam spot's diameter is approximately 100 μm, which is much larger than the domain size of bulk MnBi at low temperatures (typically $10^0$–$10^1$ μm). Therefore, the data are average values for the Fermi surfaces from different domains with the in-plane magnetic spin structure, showing a six-fold symmetry.

The ARPES intensity plots along $M_1$–$\Gamma$–$M_4$ and $M_2$–$M_3$ are shown in Fig. 4c, d and Supplementary Fig. 5. The $k_z$-dispersion is also shown in Supplementary Fig. 6. The band structure was calculated by considering a rigid energy shift overlapped with the ARPES spectrum. This calculation is consistent with the experimental data. The observed electron bands are fitted with a parabola. Consequently, we find that the Fermi velocity is approximately 3.6–10.2 eVÅ. The effective mass is in the range of 0.42–2.54 $m_0$ ($m_0$ is the mass of an electron) due to the dispersive band structure around the Fermi energy $E_F$. The detailed fitting is shown in Supplementary Fig. 7. This result confirms the previous calculation[26], and the value is smaller than 0.98–3.9 $m_0$ in the magnetic Weyl semi-metal $Co_3Sn_2S_2$[1]. These bands are responsible for the low-frequency oscillations shown in the inset of Fig. 1c[26]. According to the relationship between $\mu$ and $m_0$, which can be expressed as $\mu = \frac{q}{m_0}\tau$, where $q$ is the elementary charge and $\tau$ is the average scattering time; the low effective mass is the origin of the high transport mobility in MnBi.

While the conventional characteristic of larger mobility is seen from trivial holes and electrons ($E\sim p^2$), here the unique contribution of the linear dispersion of the topologically non-trivial bands ($E\sim p^1$) are complementary to the electron–hole pockets. Furthermore, the unique case topological character of the bands in magnetic MnBi, as shown in Supplementary Fig. 5, allow for the reduction of backscattering which would only increase the mobility. While the linear crossing is present in other ferromagnetic topological metals, MnBi is "special" in that the Bi 6$p$ bands contribute to the magnetism[27] in addition to the complementary connection between the electron–hole

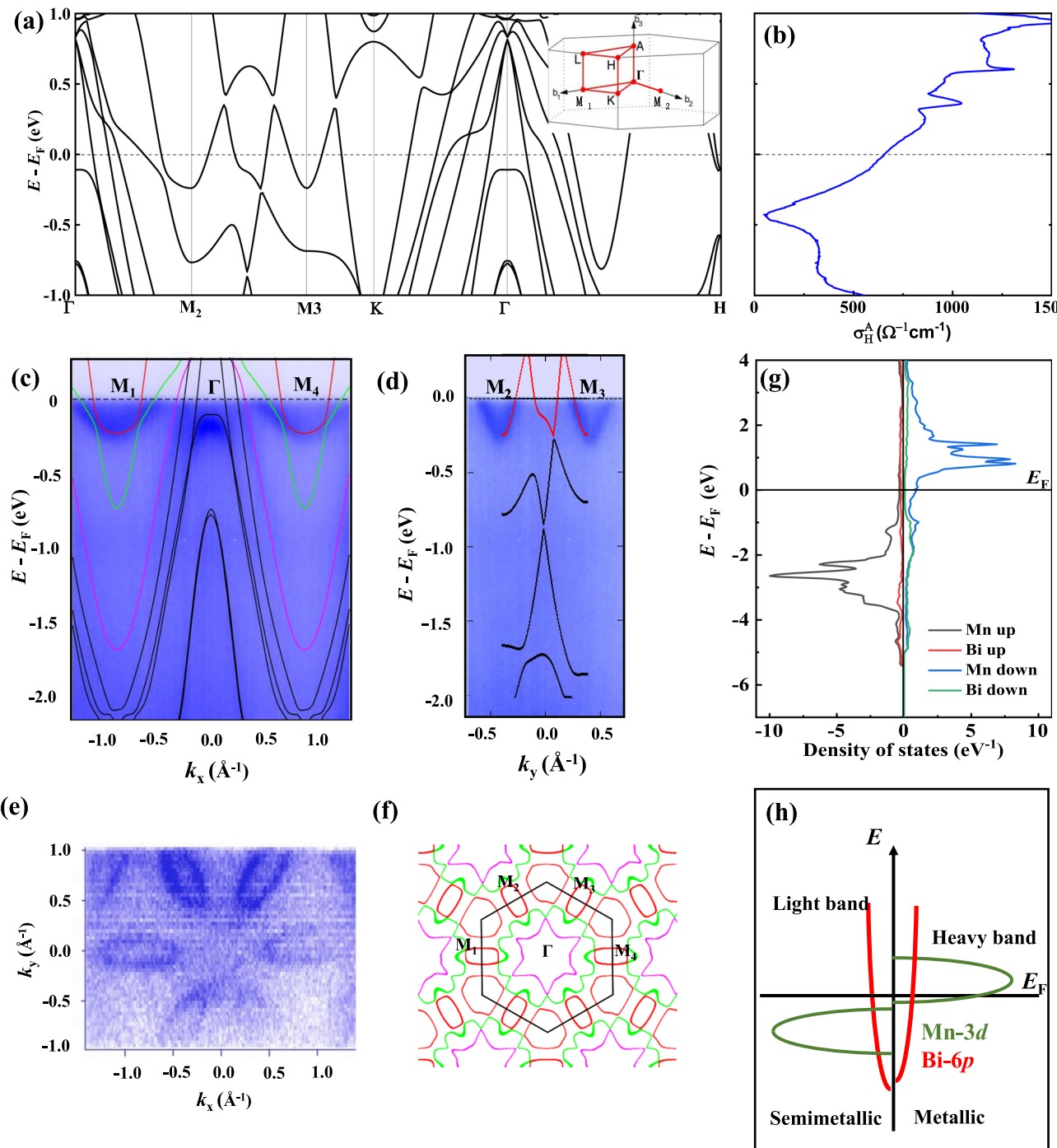

**Fig. 4 Electronic structure of MnBi. a** Band structure with the magnetic moment parallel to the *a*-axis with spin–orbit coupling. The inset shows the Brillion zone of the hexagonal structure. **b** Intrinsic anomalous Hall conductivity calculated from the Berry curvature from the band structure. **c** ARPES intensity plot along with the $M_1$–$\Gamma$–$M_4$ direction with calculated bands overlaid on top. **d** ARPES intensity plot along with the $M_2$–$M_3$ direction with calculated bands overlaid on top. **e** Fermi surface map of MnBi at $k_z = 0$, which represents the photoemission intensity integrated over a small energy window $E = E_F \pm 20$ meV around the Fermi level. **f** Calculated Fermi surface at $k_z = 0$. **g** Density of states. **h** Schematic view of the density of states, showing light semi-metallic bands in the spin-up channel and heavy metal bands in the spin-down channel.

compensation and the topological bands. The linear crossing leads to a special case of electron–hole compensation that shows to drastically enhances the mobility of magnetic systems.

## Discussion

The high mobility can also be explained by states' density, as shown in Fig. 4g, h. According to our calculation and previous reports[26,28], the spin moment for Mn is 3.7 μB, and the orbital moment is 0.2 μB. In contrast, the rest moment of approximately 0.1 μB is contributed by the polarized Bi. Whereas Mn 3*d* electrons are more localized and do not conduct electricity well, 6*p* electrons in Bi are delocalized, thus dominating the transport properties. Meanwhile, the spin polarization of MnBi is high[29], indicating that there is no large Mn 3*d* density of states at the Fermi energy[26,28] in the spin-up channel, with the fully occupied

Mn $3d^5$ approximately 2 eV below the Fermi energy. All the electrons mentioned above and magnetic structures indicate MnBi as a magnetic "Bi" from transport properties.

Bi and several of its alloys and compounds, such as Bi–Sb[30], LaBi[31], GdPtBi[32], and $Bi_2Te_3$[33], are topological materials with high mobility, owing to the highly delocalized 6p electrons of Bi and its high spin–orbital coupling. In MnBi, the Bi 6p orbital is strongly hybridized with Mn 3d electrons, confirmed by the induced magnetism in Bi, as proven using X-ray magnetic circular dichroism[27]. Therefore, the transport properties are dominated by the magnetic Bi 6p electrons. Despite the record mobility and large intrinsic anomalous Hall conductivity, MnBi also shows a large anomalous Nernst effect due to the intrinsic Berry curvature[34], which further confirms the combination of magnetism and topological band structure and the novelty of adding Mn and Bi together.

Mn is often used as a magnetic element in topological materials, such as $Mn_3Sn$[35] and $EuMnBi_2$[36], where a large AHE and quantum Hall effect have been realized. Unfortunately, the Mn atoms' magnetic moments in these materials are antiferromagnetically coupled, either non-collinearly or collinearly. To achieve ferromagnetism for realizing the quantum AHE, the thickness of the two-dimensional anti-ferromagnet $MnBi_2Te_4$ must be reduced to a few nanometers so that an even number of layers can achieve ferrimagnetism[37]. So far, ferromagnetic Weyl semi-metals are mainly Co-based, e.g., $Co_2MnGa$[2,9] and $Co_3Sn_2S_2$[1], with small moments of 1.33 (which is also mainly provided by Mn) and 0.3 μB per magnetic atom, respectively. MnBi has a relatively large magnetic moment of 3.94 $\mu_B$/Mn, one of the largest in a rare-earth-free ferromagnetic material. A large moment can be easily detected and influenced by the magnetic field. The simple crystal structure of alternative Mn and Bi layers, combined with a high Curie temperature (630 K), makes MnBi a good candidate for thin films for applications in future spintronics as a sensor for detecting temperature, field, and orientation.

In summary, MnBi exhibits abnormal transport properties. Its high mobility of 5000 $cm^2\,V^{-1}\,s^{-1}$ at 2 K is the highest for ferromagnetic materials. A positive linear MR of 5000% is not saturated up to 70 T. The two-charge-carrier behavior with a relatively low density, together with an intrinsic $\sigma_H$ of 450 $\Omega^{-1}\,cm^{-1}$, indicates a topological band structure in the momentum space, making MnBi similar to magnetized Bi. The effective mass is as small as 0.42 $m_0$. The linear MR is strongly temperature-, field-, and orientation-dependent, indicating that MnBi might be suitable for use as a sensor in future spintronics.

## Methods

**Single crystal growth**. High-quality MnBi crystals were grown using the flux method. Accordingly, 0.36 g of Mn (99.95%) and 15 g of Bi (99.9999%) pieces were ground lightly, mixed in an alumina crucible, and sealed in an evacuated quartz tube. The tube was heated to 1273 K for 20 h, held for 24 h, cooled to 653 K in 10 h, and then slowly cooled to 548 K within 170 h before centrifugation. Short bar-like single crystals of size a few millimeters were formed. The crystals were carefully polished to remove any remaining Bi flux from the surface. Subsequently, they were examined using scanning electron microscopy to confirm the clean surface. MnBi is not stable in air, and the crystals will be partly oxidized after being exposed to air overnight. Therefore, single crystals were used for the experiments immediately after cleaning the surface.

MnBi single crystals were confirmed with a NiAs-type crystal structure via powder X-ray diffraction using a monochromator's Cu Kα radiation. The composition was characterized using energy-dispersive X-ray (EDX) analysis to be $Mn_{49.2}Bi_{50.8}$ (uncertainty in EDX is 1%). The orientation of the single crystals was confirmed using the Laue method.

**Magnetization**. The magnetization measurements were conducted on single crystals with the magnetic field applied along either the a- or c-axis using a vibrating sample magnetometer (MPMS 3, Quantum Design). The sample was carefully immobilized with glue considering the strong torque. At 300 K, MnBi is a good hard magnet with large magnetostriction, and the magnetization process along the hard axis at 9 T can break the bulk single crystal.

**Transport properties**. The longitudinal and Hall resistivities were measured using a Quantum Design PPMS 9 with a standard four-probe method. The accuracy of the resistivity measurements was ±5%. For room-temperature transport measurements, single crystals were cut to $17.10 \times 2.24 \times 1.16$ μm³ using a focused ion beam to fix the samples without breaking them. High-field MR measurements were performed at the Dresden High Magnetic Field Laboratory using a pulsed magnet of 70 T.

**First-principles calculations**. We utilize first-principles calculations in the full-potential linearized augmented plane-wave code fleur (See http://www.flapw.de). Here, we use a plane-wave cut-off of 3.7 a.u.$^{-1}$ and a k-mesh of $16 \times 16 \times 7$ in the Brillouin zone. To reproduce the electronic structure and anisotropy effects[26] observed in experiments accurately, we utilize the GGA+U parameterization with $U = 4.0$ eV and $J = 0.97$ eV at the Mn atoms.

**ARPES Measurements**. Soft X-ray-ARPES measurements were performed at the SX-ARPES endstation[38] of the ADRESS beamline[39] at the Swiss Light Source, Paul Scherrer Institute, Switzerland, using a SPECS analyzer. The increase of photoelectron means free path in the soft-X-ray energy range results, by the Heisenberg uncertainty principle, in the high intrinsic resolution of the ARPES experiment in the out-of-plane momentum $k_z$[40], allowing us to precisely locate electron states in the 3D Brillouin zone. The sample was cleaved in situ at 15 K with a base pressure lower than $1 \times 10^{-10}$ mbar. The data were collected using circularly polarized light with an overall energy resolution of 50–80 meV. The photon energy is in the soft X-ray region (300–800 eV).

## Data availability

The datasets generated and/or analyzed during the current study are available from the corresponding author on reasonable request.

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

## Acknowledgements

This study was financially supported by an Advanced Grant from the European Research Council (No. 742068) "TOPMAT", the European Union's Horizon 2020 research and innovation program (No. 824123) "SKYTOP", the European Union's Horizon 2020 research and innovation program (No. 766566) "ASPIN", the Deutsche Forschungsgemeinschaft (Project-ID 258499086) "SFB 1143", the Deutsche Forschungsgemeinschaft (Project-ID FE 633/30-1) "SPP Skyrmions", and the DFG through the Würzburg-Dresden Cluster of Excellence on Complexity and Topology in Quantum Matter ct.qmat (EXC 2147, Project-ID 39085490). We acknowledge the support of the Dresden High Magnetic Field Laboratory (HLD) at HZDR and members of the European Magnetic Field Laboratory (EMFL).

## Author contributions

Y.H. and C.F. conceived this work. The single crystals were grown by Y.H. The crystal, magnetic, and transport measurements were characterized by Y.H. with the help of T.H., T.R., W.S., and M.N. The FIB microstructure transport devices were fabricated by T.H., J.G., and Y.S., and G.H.F. provided the theoretical support. The ARPES measurements were conducted by M.Y. and supported by V.N.S. All the authors discussed the results. The manuscript was written by Y.H. and G.H.F., with feedback from all the authors. The project was supervised by C.F.

## Funding

## Competing interests

The authors declare no competing interests.
