## [Peer Review File · Nature Communications]

Reviewers' Comments:

Reviewer #1:

Remarks to the Author:

The authors report a very large and non-saturating MR in MnBi, which they clearly show to originate in spin-polarized, sharply dispersive Bi-6p bands near E_F . Binary Mn pnictides have a big five years since the discovery of superconductivity in MnP, and the present work is no exception. I found the results to be clear, compelling, and highly interesting. Naively, one would expect MnBi to be a simple Mn^{3+} system with 4 localized d electrons in a high spin configuration (which is more or less true), but it is easy to forget about the delocalized Bi electrons, which evidently lead to the properties reported here. The observation of the highest μ in FM materials alone is an outstanding result. I recommend the work be published in Nature Communications after a few very minor adjustments. Here are some specific comments:

- The introduction includes plenty of clear explanation -- even so far as the very definition of MR -- making the topic accessible to non-experts, which is appropriate in a multidisciplinary journal like Nature Communications. Similarly, I appreciated the small, extra details interspersed throughout the manuscript, for example printing the specific model used to fit the Hall resistivity and the very clear explanation as to why the AHE is very small. Again, these details will make the work accessible both to students and non-experts.
- My only semi-substantial complaint is that fonts on all the insets in figures 1, 3, and S3 are far too small to read, even if these figures are intended to be printed across two columns. These font sizes must be increased substantially.
- I found it shocking that the large MR reported in figures 2(c) and 2(d) had not previously been reported, at least up to an accessible field like 9-14 T. I suppose the novelty is in part a testament to the difficulty in working with MnBi, which is apparently unstable in air.
- Figure 2 is only briefly discussed, as is appropriate, but I appreciate that it represents a comprehensive and huge amount of work and data.
- The author's tactic of splitting MR and Hall effect data in low temperature (<40 K) and high temperature (>40 K) in figures 2 and 3 is effective and contributes to the narrative.
- In the methods section, did the authors take any steps to investigate the possibility of Bi flux inclusions within their crystals? Were any observed after cleaving samples for ARPES measurements? These details should be included. I recognize that Bi inclusions would not be able to explain the observed MR. Along these lines, what is the experimental uncertainty in the composition as determined by EDX? I did notice that stoichiometric occupancy was sufficient to model the powder XRD pattern in the SI, so one might conclude that any excess Bi in the EDX composition is related to a contaminant.
- Finally, for those unfamiliar with the details of performing FLAPW calculations, does the J value refer to Hund's J or the exchange?

Reviewer #2:

None

Reviewer #3:

Remarks to the Author:

This manuscript reports results from a careful magneto-transport study of the interesting ferromagnet (and permanent magnet) MnBi. The main finding is a large and non-saturating linear magnetoresistance at low temperature. The authors conclude that this arises from high mobility electron and hole bands. There are several issues that require clarification before this manuscript should be published, listed below. I think points 1 and 3 are particularly relevant in determining the suitability of this manuscript for Nature Communications.

1) The connection between MnBi and topological materials needs to be better stated. There are some parallels drawn in a few places, but I am left with the conclusion that MnBi is probably not topological. nontrivial. However, this is not stated clearly and I am left a bit puzzled.

Here are a couple of examples in the text that suggest MnBi is being compared to topological semimetals but not quite saying that MnBi is a topological semimetal:

"Topological materials show a large positive MR with high mobilities due to the topological bands

and small effective masses. MnBi has the highest mobility among ferromagnetic materials.”
“In this sense, MnBi with a MR comparable to nonmagnetic semimetals, is unique in comparison to the topological ferromagnetic counterparts.”
The paper contains some first principles calculations. Could they be used to determine the potential topological nature of MnBi?

2) Pearson’s crystal structure database shows that MnBi has an orthorhombic structure at low temperatures, below the spin reorientation. It is in the authors Ref 19. This must be made clear in the text. Even if the small distortion is ignored in the analysis, it should be recognized as being ignored.

3) The connection between the transport and the electronic structure needs to be explained a little more clearly and consistently. I first understood the large MR to be due to linearly dispersing band at the fermi level:

“In other words, our results indicate that there exists an approximately linear energy spectrum in the band structure and carriers of a very low effective mass.”

“These phenomena are due to the linear dispersion of the 6p band at the Fermi energy with a small effective mass...”

But the description of the ARPES results seems to attribute the transport behavior to two separate electron and hole pockets:

“A series of hole pockets located at the G point is surrounded by six electron pockets located at the M points, confirming the two-charge-carrier behaviour in transport measurements.”

4) I found the presentation of the MR data in Figure 1c to be misleading. Showing ρ vs B that has oscillations throughout and especially at high field, with an inset showing SdH oscillations in $1/B$ suggests strongly that the two oscillations are related. But the text says that the wiggles in $r(B)$ actually arise from fluctuations in B in the high field experiment. I suggest the authors show only sparse data from the high field run, every 5T or so, to eliminate this (likely unintentional) misleading of the reader. The full, noisy data could be shown in the supplemental if properly captioned.

There is a topographical error in the y-axis title of Figure S1. It should be $\rho(T)/\rho(2K)$.

Reviewer #4:

Remarks to the Author:

A large non-saturating magnetoresistance, often originating from the electron-hole compensation and/or topological electronic structure with linear band crossings, has prime importance in both fundamental physics and technological applications. The large MR in magnetic material is particularly interesting, as the spin degrees of freedom provides an additional tuning knob to control the MR. In the present manuscript, He et al. has reported the large non-saturating MR as well as the high mobility in ferromagnetic MnBi. The large MR in magnetic materials has been reported since 1990s. To warrant the publication in Nature Communications, I hope the authors to clarify the points below related to the novelty of the manuscript.

1) Does the topological band structure of MnBi have any relevance to the observed large MR? In case of the Weyl semimetals the authors have referred in the introduction, the topological electronic structure with linear band crossing has direct consequence in the large MR and high mobility. In contrast, the large MR and high mobility in MnBi seem to originate from the electron-hole compensation in sharp Bi 6p bands which is rather conventional mechanism.

2) The authors measured MR up to impressively high field regime ~ 70 T to observe large non-saturating MR of 5000 %. For a fair comparison with other materials, this is equivalent to 1000 % at 14 T, which is 1/500 times smaller than other well-known large MR materials, for example, WTe₂. Instead of comparing MR in MnBi with other large MR materials, in Fig. 1e the authors present its comparison with other topological magnets, which are not particularly known for the large MR behaviors. Considering that the topological band structure of MnBi does not bear direct relevance to its MR (see 1), I am not sure what is the point of comparing MR in MnBi with other

topological magnets. More insights would be obtained if the authors present its comparison with other large MR materials (either magnetic or nonmagnetic).

3) The authors mentioned that they pinpoint the $k_z = 0$ of the Brillouin zone from the photon-energy dependent soft X-ray ARPES experiments. To justify this point, I believe it is important to include the experimental k_z -dispersion somewhere in the manuscript.

REVIEWER COMMENTS

Reviewer #1 (Remarks to the Author):

The authors report a very large and non-saturating MR in MnBi, which they clearly show to originate in spin-polarized, sharply dispersive Bi-6p bands near E_F . Binary Mn pnictides have a big five years since the discovery of superconductivity in MnP, and the present work is no exception. I found the results to be clear, compelling, and highly interesting. Naively, one would expect MnBi to be a simple Mn^{3+} system with 4 localized d electrons in a high spin configuration (which is more or less true), but it is easy to forget about the delocalized Bi electrons, which evidently lead to the properties reported here. The observation of the highest μ in FM materials alone is an outstanding result. I recommend the work be published in Nature Communications after a few very minor adjustments. Here are some specific comments:

Thanks a lot for supporting our paper.

- The introduction includes plenty of clear explanation -- even so far as the very definition of MR -- making the topic accessible to non-experts, which is appropriate in a multidisciplinary journal like Nature Communications. Similarly, I appreciated the small, extra details interspersed throughout the manuscript, for example printing the specific model used to fit the Hall resistivity and the very clear explanation as to why the AHE is very small. Again, these details will make the work accessible both to students and non-experts.

Thanks a lot for supporting our paper.

- My only semi-substantial complaint is that fonts on all the insets in figures 1, 3, and S3 are far too small to read, even if these figures are intended to be printed across two columns. These font sizes must be increased substantially.

We rearrange the fonts in the insets as required.

- I found it shocking that the large MR reported in figures 2(c) and 2(d) had not previously been reported, at least up to an accessible field like 9-14 T. I suppose the novelty is in part a testament to the difficulty in working with MnBi, which is apparently unstable in air.

The most difficult thing is to grow single crystals for MnBi. Simply cooling down from the melt cannot lead to hexagonal MnBi, but a mixture of several phases. It can be grown only by the flux method, rather than conventional methods like Bridgeman. The air stability is not that bad, but care has to be taken.

- Figure 2 is only briefly discussed, as is appropriate, but I appreciate that it represents a comprehensive and huge amount of work and data.

Thanks a lot for supporting our paper.

- The author's tactic of splitting MR and Hall effect data in low temperature (<40 K) and

high temperature (>40 K) in figures 2 and 3 is effective and contributes to the narrative.

Thanks a lot.

- In the methods section, did the authors take any steps to investigate the possibility of Bi flux inclusions within their crystals? Were any observed after cleaving samples for ARPES measurements? These details should be included. I recognize that Bi inclusions would not be able to explain the observed MR. Along these lines, what is the experimental uncertainty in the composition as determined by EDX? I did notice that stoichiometric occupancy was sufficient to model the powder XRD pattern in the SI, so one might conclude that any excess Bi in the EDX composition is related to a contaminant.

We notice the possibility of having Bi flux inside the crystals. To avoid this, we selected the crystals carefully. The crystals were polished to remove the flux on the surface and to reduce their size so that there is less chance to have the flux inside the crystals. Furthermore, crystals cut by the focused ion beam technique were used for some measurements. Those crystals were as small as $1\ \mu\text{m}$, which is free from Bi checked by both sides. We did not observe Bi after cleaving in ARPES measurements. The uncertainty in the EDX is 1%, which could not be used to determine the Bi contamination.

- Finally, for those unfamiliar with the details of performing FLAPW calculations, does the J value refer to Hund's J or the exchange?

The J is the non-spherical part of the Coulomb interaction and U is the spherical part. See Solovyev and Dederichs, Phys. Rev. B 49, 6736 (1994).

Reviewer #3 (Remarks to the Author):

This manuscript reports results from a careful magneto-transport study of the interesting ferromagnet (and permanent magnet) MnBi. The main finding is a large and non-saturating linear magnetoresistance at low temperature. The authors conclude that this arises from high mobility electron and hole bands. There are several issues that require clarification before this manuscript should be published, listed below. I think points 1 and 3 are particularly relevant in determining the suitability of this manuscript for Nature Communications.

1) The connection between MnBi and topological materials needs to be better stated. There are some parallels drawn in a few places, but I am left with the conclusion that MnBi is probably not topological nontrivial. However, this is not stated clearly and I am left a bit puzzled.

Here are a couple of examples in the text that suggest MnBi is being compared to topological semimetals but not quite saying that MnBi is a topological semimetal:

“Topological materials show a large positive MR with high mobilities due to the topological bands and small effective masses. MnBi has the highest mobility among ferromagnetic materials.”

“In this sense, MnBi with a MR comparable to nonmagnetic semimetals, is unique in comparison to the topological ferromagnetic counterparts.”

The paper contains some first principles calculations. Could they be used to determine the potential topological nature of MnBi?

The topological nature of MnBi is shown below. We add a section in the supplementary information.

The band near the Fermi level on the cut between M_2 and M_3 is dominated by the spin up channel, as presented in Fig. S5. A band inversion occurs at 0.25 eV below the Fermi level. Because of the mirror symmetry $\{m_{001}|0, 0, 1/2\}$, the bands in the mirror symmetry invariant plane with $k_z=0$ can have the mirror eigenvalues 1 and -1. If two inverted bands have opposite mirror eigenvalues, a nodal line linear crossing can be formed in the $k_z=0$ plane. Indeed, the two linear crossing points between M_2 and M_3 are just locating on such nodal lines. The mirror symmetry can be broken by the spin-orbit coupling (SOC) and the applied magnetic field (not perpendicular to the mirror plane). The nodal line linear band crossing can be broken by opening a bandgap with one pair of Weyl points locating on the original nodal lines, similar to the case in $\text{Co}_3\text{Sn}_2\text{S}_2$ (Nature Physics 14, 1125, (2018)). This bandgap can have strong local Berry curvature around the mirror plane and leads to a strong anomalous Hall effect.

Figure S5. Topological nature of MnBi. (a) Calculated band structure with ARPES experiment. The red and black lines are corresponding to bands with and without SOC, respectively. Nodal lines and Weyl points are marked in yellow circles. (b) Schematic view for the band inversion. (c) Calculated band structure with the moment along (100) direction. Magnetic moment breaks the six-fold symmetry, forming nodal lines and Weyl points depending on the moment direction. It is a pity that MnBi has easy-plane magnetic structure, so that the ARPES result is an average of hundreds of magnetic domains with different magnetization directions. Therefore, it is difficult to observe clearly nodal lines and Weyl points by experiment.

2) Pearson's crystal structure database shows that MnBi has an orthorhombic structure at low temperatures, below the spin reorientation. It is in the authors Ref 19. This must be made clear in the text. Even if the small distortion is ignored in the analysis, it should be recognized as being ignored.

We add a few sentences in the revised paper to state this distortion:

Strictly speaking, the crystal structure is orthorhombic below T_{SR2} due to the magnetostriction, but this small distortion is ignored in the analysis.

3) The connection between the transport and the electronic structure needs to be explained a little more clearly and consistently. I first understood the large MR to be due to linearly dispersing band at the fermi level:

“In other words, our results indicate that there exists an approximately linear energy spectrum in the band structure and carriers of a very low effective mass.”

“These phenomena are due to the linear dispersion of the 6p band at the Fermi energy with a small effective mass...”

But the description of the ARPES results seems to attribute the transport behavior to two separate electron and hole pockets:

“A series of hole pockets located at the G point is surrounded by six electron pockets located at the M points, confirming the two-charge-carrier behaviour in transport measurements.”

The coexistence of linearly dispersing bands (topologically non-trivial $E \sim p^1$) and trivial $E \sim p^2$ holes and electrons is predicted by calculations and confirmed by the ARPES experiment. While the conventional characteristic of larger mobility is seen from trivial holes and electrons ($E \sim p^2$), here the unique contribution of the linear dispersion of the topologically non-trivial bands ($E \sim p^1$) are complementary to the electron-hole pockets. Furthermore, the unique case topological character of the bands in magnetic MnBi, as shown in Fig. S5, allow for the reduction of backscattering which would only increase the mobility. While the linear crossing is present in other ferromagnetic topological metals, MnBi is ‘special’ in that the Bi 6p bands contribute to the magnetism (Phys. Rev. B 94, 184433 (2016)) in addition to the complementary connection between the electron hole compensation and the topological bands. The linear crossing leads to a special case of electron hole compensation that shows to drastically enhance the mobility of magnetic systems.

We write an additional paragraph to reveal the role of the linearly dispersing band.

4) I found the presentation of the MR data in Figure 1c to be misleading. Showing ρ vs B that has oscillations throughout and especially at high field, with an inset showing SdH oscillations in $1/B$ suggests strongly that the two oscillations are related. But the text says that the wiggles in $r(B)$ actually arise from fluctuations in B in the high field experiment. I suggest the authors show only sparse data from the high field run, every 5T or so, to eliminate this (likely unintentional) misleading of the reader. The full, noisy data could be shown in the supplemental if properly captioned.

We change the Figure as suggested.

There is a topographical error in the y-axis title of Figure S1. It should be $\rho(T)/\rho(2K)$.

We correct the typo as suggested.

Reviewer #4 (Remarks to the Author):

A large non-saturating magnetoresistance, often originating from the electron-hole

compensation and/or topological electronic structure with linear band crossings, has prime importance in both fundamental physics and technological applications. The large MR in magnetic material is particularly interesting, as the spin degrees of freedom provides an additional tuning knob to control the MR. In the present manuscript, He et al. has reported the large non-saturating MR as well as the high mobility in ferromagnetic MnBi. The large MR in magnetic materials has been reported since 1990s. To warrant the publication in Nature Communications, I hope the authors to clarify the points below related to the novelty of the manuscript.

1) Does the topological band structure of MnBi have any relevance to the observed large MR? In case of the Weyl semimetals the authors have referred in the introduction, the topological electronic structure with linear band crossing has direct consequence in the large MR and high mobility. In contrast, the large MR and high mobility in MnBi seem to originate from the electron-hole compensation in sharp Bi 6p bands which is rather conventional mechanism.

While the conventional characteristic of larger mobility is seen from trivial holes and electrons ($E \sim p^2$), here the unique contribution of the linear dispersion of the topologically non-trivial bands ($E \sim p^1$) are complementary to the electron-hole pockets. Furthermore, the unique case topological character of the bands in magnetic MnBi, as shown in Fig. S5, allow for the reduction of backscattering which would only increase the mobility. While the linear crossing is present in other ferromagnetic topological metals, MnBi is “special” in that the Bi 6p bands contribute to the magnetism (Phys. Rev. B 94, 184433 (2016)) in addition to the complementary connection between the electron hole compensation and the topological bands. The linear crossing leads to a special case of electron hole compensation that shows to drastically enhance the mobility of magnetic systems.

We write an additional paragraph to reveal the role of the linearly dispersing band.

2) The authors measured MR up to impressively high field regime ~ 70 T to observe large non-saturating MR of 5000 %. For a fair comparison with other materials, this is equivalent to 1000 % at 14 T, which is 1/500 times smaller than other well-known large MR materials, for example, WTe₂. Instead of comparing MR in MnBi with other large MR materials, in Fig. 1e the authors present its comparison with other topological magnets, which are not particularly known for the large MR behaviors. Considering that the topological band structure of MnBi does not bear direct relevance to its MR (see 1), I am not sure what is the point of comparing MR in MnBi with other topological magnets. More insights would be obtained if the authors present its comparison with other large MR materials (either magnetic or nonmagnetic).

Ferromagnetic materials usually have a large density of states at the Fermi level. The interaction from electrons greatly decreases the mobility. In addition, the spin scattering effect, which is absence in nonmagnetic systems, also significantly limits the mobility. Therefore, ferromagnetic systems usually have a much lower MR than nonmagnetic systems. It is unfair to compare nonmagnetic systems with magnetic systems where

additional mechanisms exist.

We add additional words in the main text to show that nonmagnetic systems have much larger MR and explain the reason.

3) The authors mentioned that they pinpoint the $k_z = 0$ of the Brillouin zone from the photon-energy dependent soft X-ray ARPES experiments. To justify this point, I believe it is important to include the experimental k_z -dispersion somewhere in the manuscript.

We have added the k_z -dispersion in the supplemental information. We also shown it here for convenience. The hole pocket at Γ point and electron pocket at M points can be clearly seen.

Figure S6. ARPES intensity plot in the k_x - k_z plane at $k_y = 0$, 48 meV below E_F , acquired with circular polarized photon with photon energy ranging from 380 to 420 eV. The M_1 - Γ - M_2 plane lies at $k_z = 10.27 \text{\AA}$.

Reviewers' Comments:

Reviewer #1:

Remarks to the Author:

The authors have appropriately responded to my previous comments. I agree with the authors that the experimental ARPES intensity plots along M2-M3 shown in figure S5(a) are less than compelling. Nonetheless, plotting these data with the calculated electronic structure while providing an explanation as to their shortcomings gives the reader an adequate opportunity to assess the strength of the authors' conclusions. I believe the manuscript can be accepted in present form.

Reviewer #3:

Remarks to the Author:

I believe that the authors have adequately addressed the concerns and comments of all three reviewers. The additions to the text, in particular relating to the topological band structure, has improved the manuscript considerably. I think the revised paper should be published.

Reviewer #4:

Remarks to the Author:

The authors provided satisfactory reply to most questions I had and modified the manuscript accordingly. This includes an extensive proof of the topological character of MnBi (Fig. S5) and an inclusion of k_z -dependent SX-ARPES data (Fig. S6)

As agreed by the authors, the topological character of MnBi plays only marginal role in terms of the origin of large MR behavior, though it enhances mobility. This makes the mechanism behind large MR in magnetic MnBi not very different from previously reported large MR materials. For example, it is not entirely clear to me why the fact that Bi 6p bands contribute to the magnetism (by very small amount 0.1 μ_B) makes it "special" in terms of the mechanism behind large MR.

However, I agree with the authors that ferromagnetic MnBi with large MR and high mobility may have technical importance in the field of spintronics.

REVIEWERS' COMMENTS

Reviewer #1 (Remarks to the Author):

The authors have appropriately responded to my previous comments. I agree with the authors that the experimental ARPES intensity plots along M2-M3 shown in figure S5(a) are less than compelling. Nonetheless, plotting these data with the calculated electronic structure while providing an explanation as to their shortcomings gives the reader an adequate opportunity to assess the strength of the authors' conclusions. I believe the manuscript can be accepted in present form.

We are grateful for the affirmation of our paper.

Reviewer #3 (Remarks to the Author):

I believe that the authors have adequately addressed the concerns and comments of all three reviewers. The additions to the text, in particular relating to the topological band structure, has improved the manuscript considerably. I think the revised paper should be published.

We are grateful for the affirmation of our paper.

Reviewer #4 (Remarks to the Author):

The authors provided satisfactory reply to most questions I had and modified the manuscript accordingly. This includes an extensive proof of the topological character of MnBi (Fig. S5) and an inclusion of k_z -dependent SX-ARPES data (Fig. S6)

As agreed by the authors, the topological character of MnBi plays only marginal role in terms of the origin of large MR behavior, though it enhances mobility. This makes the mechanism behind large MR in magnetic MnBi not very different from previously reported large MR materials. For example, it is not entirely clear to me why the fact that Bi 6p bands contribute to the magnetism (by very small amount 0.1 uB) makes it "special" in terms of the mechanism behind large MR.

Ferromagnetic topological materials are of special interest for condensed matter physics. Bi is a typical element that usually supports the topology. Ferromagnetism is always desired to break the time-reversal symmetry in topological materials. Despite the dilute systems, such as Cr-doped Bi_2Te_3 , the ferromagnetism in MnBi is intrinsic, which makes it special.

However, I agree with the authors that ferromagnetic MnBi with large MR and high mobility may have technical importance in the field of spintronics.

We are grateful for the affirmation of our paper.